# Multilingual Research Writing beyond English: The Case of Norwegian Academic Discourse in an Era of Multilingual Publication Practices

**Kristin Solli \* and Ingjerd Legreid Ødemark**

OsloMet—Oslo Metropolitan University, 0130 Oslo, Norway; ingjerd-legreid.odemark@oslomet.no
\* Correspondence: kristin.solli@oslomet.no; Tel.: +47-4760-7121

**Abstract:** Although English is the dominant language of scholarly publication, many multilingual scholars continue to publish in other languages while they also publish in English. A large body of research documents how these multilingual scholars negotiate writing in English for publication. We know less, however, about the implications of such negotiations for other languages that scholars work in. We wanted to investigate trends in writing conventions in language other than English during a period when multilingual publication patterns have been common. Specifically, we examined changes in rhetorical patterns in the introduction sections of the 1994 and the 2014 volumes of three Norwegian-language journals in three different disciplines in the humanities and social sciences. Our findings show that while certain features of our material might be interpreted as the result of a non-English discourse community adopting dominant Anglo-American models, the overall picture is more complex. Our study indicates that we need more research that examines cross-linguistic textual practices that focus on English and any other languages that scholars may work in. We also consider the possible pedagogical implications of such a focus.

**Keywords:** multilingual scholars; writing for publication; diachronic studies; introduction sections; rhetorical analysis; Norwegian academic discourse

## 1. Introduction

### 1.1. Why Is a Focus on Other Languages than English Needed?

Although English is the dominant language of scholarly publication, many multilingual scholars continue to publish in other languages while they also publish in English [1–3]. A large and important body of research documents how multilingual scholars negotiate writing in English for publication [3–12]. This work has been crucial to highlight the pressures, dilemmas and challenges multilingual scholars face when they want to publish in English. This literature has, however, had less to say about the other languages that multilingual scholars work in. That is not to say, of course, that other languages are not present in these studies but they tend to function as a backdrop to explain why writing in English is a challenge rather than as a focal point. As such, other languages sometimes appear as rather static entities against which English is contrasted.

In other words, despite a flourishing interest in the texts and writing practices of multilingual scholars, most of this research has tended to put English at the centre of attention. Fewer studies have examined the implications of multilingual publishing practices for academic discourses in languages other than English or at least such studies are less often available in English-language journals. This tendency to put English at the centre of research about multilingual scholarly writing is on the one hand both reasonable and necessary because it is crucial to understand the role of English in the

global production of knowledge. On the other hand, it is also problematic because this focus can render other languages invisible or peripheral, as if all multilingual researchers only write and publish in English. For a richer understanding of the implications of multilingual publication practices, it is important to understand possible implications of such practices for other languages than English as well.

We thus wanted to find out more about trends in writing conventions in another language than English during a period when multilingual publication patterns have been common. Specifically, we examined changes in rhetorical patterns in the introduction sections of three Norwegian-language journals in three different disciplines in the humanities and social sciences: education, sociology and literary studies. Because we wanted to investigate potential changes during a time of increased pressure to publish in English for Norwegian researchers, we analysed the introduction sections of the 1994 and 2014 volumes of the journals.

Our findings show that during this period it has become more common to include explicit statements of positioning in relation to previous research and statements declaring the purpose of the article. These changes suggest that rhetorical conventions that are common in Anglo-American academic discourse are becoming more common in Norwegian discourse as well. However, our analysis highlights that the role of English is only one of several possible explanations for this trend. Moreover, another striking feature in both introductions from 1994 and those from 2014 is the great variation in the overall rhetorical organization, complicating indications of increased standardization. In other words, our material does not allow us to draw firm conclusions about the potential influence of English language dominance on Norwegian academic discourse. More than anything, our study highlights the intricate and complex relationships between languages, disciplines and cultures.

As such, our study identifies more questions than it answers. However, we believe that such questions should be of interest to researchers and teachers of research writing in multilingual contexts. Even though English is the lingua franca of research, it is not the only language of research and we need studies that attend to writing conventions in other languages as well. As we discuss in more detail in our concluding section, such studies are particularly important given the dominance of English. Specifically, we argue for the importance of studies that examine cross-linguistic textual practices with an eye towards change, flux and complexity and for studies that attempt to understand writing conventions within the particular historical, social and political contexts in which they occur. Finally, we also consider some pedagogical implications such a focus might have and we join those who have pointed to the limitations of the current dominant practice of monolingual teaching and learning of academic writing [3,13–16].

## 1.2. Multilingual Writing Practices and Shifting Writing Conventions: Language, Discipline, Culture

In order to place our study within current research on multilingual research writing, this section outlines some of the central approaches to studying multilingual writing practices and changing writing conventions that this study draws on. Our understanding of the term "multilingual" builds on current research that uses the term to describe scholars who write for publication in English but who are "working and living in contexts where English is not the official or dominant means of communication" [3] (p. 1). There have been numerous influential studies of the writing practices of such scholars and several of them have examined the dynamic between English and other languages. The work of Teresa Lillis and Mary Jane Curry [3], for example, has focused on how multilingual researchers make decisions about what research topics multilingual scholars deem appropriate for publication in their first language or other languages and which they deem appropriate to publish in English. While Lillis and Curry do consider how multilingual scholars negotiate research in relation to several different research and linguistic communities, they do not address how the pressures to publish in English might influence how these scholars approach writing in their first languages at a textual level.

In terms of work that focuses on textual analysis of research writing, the field of intercultural rhetoric and its predecessor, contrastive rhetoric, has a long history of studying the national and

cultural specificities of academic discourse in different languages. In many ways, intercultural rhetoric serves as an important starting point for our study. For example, our project assumes that there is such a thing as a "Norwegian" academic writing tradition and an "Anglo-American" academic writing tradition. This is in line with the central claims of traditional intercultural rhetoric [17–19]. As Diane Belcher points out, many scholars in the field of intercultural rhetoric have since questioned the "monolithic cultural determinism" that characterizes some of the earlier work in field [20] (p. 64).

And indeed, corpus-based studies have complicated the idea of such determinism. For example, Fløttum et al.'s project "Cultural Identity in Academic Prose" analysed research articles in three disciplines (economics, linguistics and medicine) across three languages: Norwegian, French and English [21]. Their findings suggest that there are greater varieties between disciplines than between languages and they conclude that neither discipline nor language can fully explain the differences they have found. Similarly, Shaw and Vassileva conducted a diachronic study of economics journals in four languages to determine whether differences are due to "essential cultural differences" or other "external" factors such as disciplinary developments or the status of languages [22] (p. 1).They find no evidence of persistent differences that can be ascribed to culture. Instead, they ascribe the changes they find to disciplinary developments, changes in the status of languages and local publishing practices and conventions.

These studies, then, reject "culture" in and of itself as an explanation for particular discursive and rhetorical styles. Bennett and Muresan [23], however, make a point of recuperating the idea of the cultural root of discourses. In particular, they wish to demonstrate the existence of a set of discursive ideals particular to "romance cultures." They do so in order to argue for the importance of valuing and bolstering local writing traditions and epistemic practices in face of the hegemony of what Bennett refers to as "English Academic Discourse" [24]. Bennett argues that this discourse is not only dominant but that it is "predatory" [25]. She uses this adjective to describe how the stylistic and discursive ideals of English academic discourse are spreading to other languages that traditionally have had different ideals. Such a spread is about more than stylistics, she insists and uses the term "epistemicide" to highlight how Anglo-American ideals are replacing other ways of writing and constructing knowledge.

In a case study that highlights one example of this process, Bennett analyses the "Anglicization" of Portuguese history writing and she argues that the writing conventions in this discipline have gone through profound changes due to the import of Anglo-American writing conventions [26]. Building on Bennett's work, Geneviève Bordet argues that French discourse is in danger of falling prey to English as well. She examines the translation of metaphors from English to French and sees translation as one potentially fruitful avenue to counter uncritical adoption of Anglo-American modes of thought and writing into French [27]. Salager-Meyer and colleagues [28], also argue that the changes they observe in Spanish medical writing from 1930 until 1995 can be attributed to Spanish scholars' engagement with English language research and the emergence of guide books and writing courses based on Anglo-American writing traditions. In sum, these studies suggest that "romance" academic discourses are undergoing profound changes due to the dominance of English. Inspired by such studies, we wanted to explore whether such trends also extend to other languages.

*1.3. Language Policies and Publication Practices in Norway*

Because this study is concerned with the historical and cultural specificity of academic writing, it is necessary to provide a brief introduction to Norwegian academic discourse and the broader context from which our material has been gathered. The point is not to provide an exhaustive account of Norwegian academic discourse but rather to highlight the processes and dynamics that might resonate in other contexts in which English is used as an additional language.

During the last several decades, the dominance of English in higher education and scholarly publications has been a cause of public worry in Norway. In a report from 2017 about the state of Norwegian language in various fields, the Norwegian Language Council describes higher education as an area in which the status of Norwegian is "threatened" and "under great pressure due to

increased demands to internationalize"[29] (p. 29). For example, the report points to a figure showing that 90% of PhD dissertations from 2013–2015 were written in English (p. 32). The Language Council say they "worry" that institutions of higher education are doing very little to actively counter this trend of English-language dominance [29] (pp. 41–42).

Despite such anxieties, current language policies recognize the importance of English but advocate "parallel language use" stating that Norwegian and English or other foreign languages should have "parallel" uses. The Norwegian Association of Higher Education Institutions, for example, calls for Norwegian institutions of higher education to formulate language policies that "promote the use of Norwegian language but in a way that ensures that English or another international language may be used when appropriate or required.

This policy, then, stresses the need for all scholars to communicate with more than one linguistic community, in a sense, encouraging all scholars to become "multilingual." Yet, it has had limited practical implications when compared to the "the publication indicator," a research funding system we describe in more detail later on in this article. The opportunities for rewards in this system are much larger for scholars who publish in English-language journals than for scholars, who publish in Norwegian-language journals. In sum, while the parallel language use policy encourages publishing in Norwegian, the research funding system offers greater rewards for publishing in English. This situation has been used as one explanation for an increase in English language publications among Norwegian researchers since the introduction of the system [30].

Language choice and publication patterns, however, are perhaps more than anything governed by disciplinary conventions and traditions. For example, many scholars do not have the option of publishing their research in other languages than English. This is particularly true in the natural sciences and medicine. If scholars in these fields are to publish at all, they must write in English. In the social sciences and humanities, which are the fields investigated in this study, the use of Norwegian as a language of publication varies according to discipline. Figures from 2010–2013 show that in economics and linguistics more than 80% of the publications written by researchers affiliated with Norwegian institutions were in English. In history, however, about half of the publications were in English [31]. Based on bibliometric analyses of Norwegian researchers, Gunnar Sivertsen argues that although the relative number of publications in English has increased from 2005 to 2011, the main trend in the social sciences and humanities is that most researchers publish in both English and Norwegian [31]. Sivertsen concludes that by 2011 "Publishing in the native language and in international languages is the normal practice for the majority of researchers in the SSH" [31] (p. 367). It is in this sense our study operationalizes multilingual publication practices. Assuming that the researchers in our corpus adhered to "normal" publication practices in SSH, it is likely that the majority of the authors represented in the 2014 corpus also published in more than one language. Moreover, our primary interest was not in documenting the trajectories of individual scholars but in examining potential changes in patterns in collective bodies of texts in a period in which the tendency to publish in English and in Norwegian has emerged as the norm.

## 2. Materials and Method

We wanted to analyse potential rhetorical changes in other languages than English in research communities in which multilingual publication patterns are common. Our more specific research question was: what changes, if any, are there in rhetorical patterns in introduction sections in research articles published in Norwegian-language journals in 1994 and in 2014? Below, we explain the construction of our corpus and the design of our study.

### 2.1. Corpus

In order to answer our research question we constructed a corpus consisting of the introduction sections of all the research articles included in the 1994 and the 2014 volumes of the following Norwegian-language journals: Norsk Pedagogisk Tidsskrift/NPT (education), Sosiologi i dag/SID (sociology) and Edda. Scandinavian Journal of Literary Research (literature). Table 1 shows the size

and construction of our corpus, and a full list of titles and authors of the articles in the corpus can be found in Appendix A.

**Table 1.** Corpus by title of journal, discipline and number of introduction sections in each volume.

| Journal Title | Discipline | Introduction Sections, No. | | |
|---|---|---|---|---|
| | | 1994 | 2014 | Total |
| Norsk pedagogisk tidsskrift (NPT) | Education | 28 | 35 | 63 |
| Sosiologi i dag (SID) | Sociology | 15 | 14 | 29 |
| Edda. Scandinavian Journal of Literary Research (EDDA) | Literature | 27 | 17 | 44 |
| Total | | 70 | 66 | 136 |

The reasons for the design of this specific corpus were that we wanted our material to contain texts intended primarily for Norwegian or Scandinavian readers. We made this choice because we wanted to examine texts in which authors presumably would not be subject to explicit expectations of conforming to writing conventions in international, English-language journals. Below, we discuss the reasons for choosing these journals in relation to editorial language policies, disciplinary conventions and historical period.

### 2.1.1. Language Policies of the Journals

All three journals state explicitly that they accept articles written in any Scandinavian language, that is, Danish, Swedish or Norwegian. Most Scandinavian-language researchers are able to read and understand all of these languages. NPT does not have an explicit language policy but all the articles in our sample were written in Norwegian or Swedish. SID states that it accepts articles written in any Scandinavian language and that the journal will "consider" publishing articles from "non-Scandinavian contributors" in English [32]. Our sample did not contain any examples of the latter but it did contain two articles that had been translated from English to Norwegian. Interestingly, Edda included articles in Swedish, Danish as well as three articles written in English. The journal seems to publish articles about any literary author, work or problem in Scandinavian languages and the articles written in English address Scandinavian topics (authors, works or problems). This indicates that the journal attracts an international research community, yet at the same time sees itself as speaking to and about a Nordic community. In other words, it would be possible to consider the journal a Scandinavian-language international journal.

In sum, both the language policies and the publication practices indicate that the journals see themselves as catering to a Norwegian or Scandinavian audience, while Edda might also be considered an international journal.

### 2.1.2. Disciplines

Based solely on the criteria that we wanted to analyse texts written in Norwegian for a Norwegian audience, we could have selected any journal published in Norway. However, we also wanted the corpus to reflect fields that address subject matter and issues that are often considered to be of particular interest to a national context. Specifically, we selected journals from the humanities and social sciences because the natural sciences have had a more uniform writing tradition over a longer period of time and have followed fairly similar international conventions that predate the historical period we are examining here [33]. The humanities and social sciences have traditionally had a stronger affiliation to national research interest and research that is national in scope [31]. Our assumption was that if we were able to see changes in rhetorical patterns in these fields, it might indicate a change in the conventions of the national and regional discourse communities.

Based on this assumption, we selected education, sociology and literary studies as disciplines we wanted to examine. Education is a field which in many ways is structured by national interest and governed by national policy, although there are of course many research areas that transcend national borders. The research presented in our sample from NPT is by and large conducted in and about Norwegian schools, contexts and policies. The research in our sample from the sociology

journal is not in subject matter primarily about Norwegian issues, yet several of the articles in both volumes we examined deal with placing larger sociological phenomena in a national context. Edda describes itself as a journal that is "Nordic in profile in that most articles will examine Nordic literature" [34]. As indicated above, the topics addressed in the articles in our corpus matched this description well. In sum, we chose these journals not just because they primarily publish Norwegian/Scandinavian-language articles but also because the subject matter has a particular national and regional interest, indicating that the intended audience is national/regional.

### 2.1.3. Historical Period

We selected 1994 and 2014 as volumes to be included in our corpus. We chose this time frame because we wanted to examine potential changes during a time in which Norwegian researchers were met with more explicit expectations to publish in international journals. In 2004, a system for monitoring publication output known as "the publication indicator" was introduced in Norway. Norwegian HE institutions receive part of their funding according to how well they perform in this system. The system measures a range of indicators, of which publication output is one. The system has introduced a way of dividing journals into "level 1" and "level 2" journals in which the latter are intended to represent the most selective journals in a given field. A publication in one of these journals gives the institution more points in the system than a publication that is published in a "level 1" journal. Among the journals rated as "level 2" there are very few Norwegian-language journals and critics have argued that the system therefore in effect favours English-language publishing.

Evaluations of this system show that one effect in the social sciences and humanities is that the number of publications in both English and in Norwegian has increased [31,35]. By examining writing conventions a decade before the implementation of the system and a decade after its implementation, we sought to capture potential changes in rhetorical patterns during a time of structural change for research funding. This structure change has ushered in stronger incentives to publish internationally and increased the number of researchers in the social sciences and humanities who publish in more than one language, most typically in English and in Norwegian.

### 2.2. Analytical Instrument: The Create-A-Research-Space Model

We limited the scope of our study to analysing the rhetorical patterns in the introduction sections of research articles. This choice was in part informed by the way introduction sections are particularly dense rhetorical moments of interaction between writers and readers [36]. Hence, if we wanted to look for possible changes in rhetorical patterns, we assumed introductions might be a rich site for analysis. It was also in part informed by the substantial amount of conceptual and empirical work that already exists about introduction sections. In other words, focusing on introduction sections would ensure that we would have extensive comparative material.

To analyse the rhetorical patterns in the introduction section, we used the "Create-a-Research-Space" (CARS) model (see Figure 1) as our analytical instrument. This model was developed by John Swales [37,38] and is a fixture in both writing pedagogy and in applied linguistics research. Based on an analysis of published research article introduction sections in several fields, Swales identified three "moves" and several "steps" within each move.

Swales calls his model an "ecological" approach and argues that the metaphors in the model indicate how introduction sections situate and make space for a particular project: "Just as plants compete for light and space, so writers of RPs [research papers] compete for acceptance and recognition," Swales and Feak explain in their canonical text book, *Academic Writing for Graduate Students: Essential Tasks and Skills* [39] (p. 328). The moves of the CARS model, they explain, is a way for authors to gain such acceptance and recognition.

This ecological metaphor and the names of the moves themselves indicate the model's emphasis on positioning: the introduction establishes a territory, carves out a "niche" in this landscape and shows how the article fills this niche. One of the main rhetorical effects of an introduction, according to this model, is to convince the reader of the value of the article in question by making explicit how

the work in the article is different from or similar to work that already exists. In this way, the writer promises the reader that he or she will learn something new by reading the article.

| MOVES | | STEPS |
|---|---|---|
| **Move 1** | **Establishing a research territory** | a. By showing that the general research area is important, central, interesting, problematic or relevant in some way |
| | | b. By introducing and reviewing items of previous research in the area |
| **Move 2** | **Establishing a niche** | By indicating a gap in the previous research or by extending previous knowledge in some way |
| **Move 3** | **Occupying the niche** | a. By outlining purposes or stating the nature of the present research |
| | | b. By listing research questions or hypotheses |
| | | c. By announcing principal findings |
| | | d. By stating the value of the present research |
| | | e. By indicating the structure of the RP |

**Figure 1.** The CARS model (adapted from [39]).

We applied this model to the introduction sections in our corpus looking for the relative absence and presence of the model's moves in our corpus. There is ample precedence for applying the model in this way. Previous studies have applied the model to texts written in Hungarian [40], Arabic [41,42], Polish [43], Thai [44,45], Spanish [46,47], Brazilian Portuguese [48], Chinese [49–51], Swedish [52] and Tamil [16]. Most of these studies have applied the model in a contrastive perspective in order to compare the rhetorical structure of texts written in English with the rhetorical structure of texts written in another language

In these studies, the CARS model holds a particular status as a rhetorical structure that is perceived as characteristic of Anglo-American rhetorical traditions. The purpose of many of these studies is to examine the differences between introductions written in English and those written in other languages in order to make pedagogical recommendations about what researchers who use English as an additional language must learn if they want to publish in English.

Unlike these studies, the primary goal of our research design was not to contrast a "Norwegian model" with an "Anglo-American model," but to use the CARS model as an instrument to make comparisons of certain features possible across time. Yet, it is also worth pointing out how the CARS model differs from or is similar to traditional Norwegian writing conventions. On the one hand, it is difficult to speak of a "Norwegian model" since little empirical work on introduction sections in Norwegian research articles exists. However, at a more general level it is possible to say that a typical feature has been the absence of a clear model. Studies of Norwegian-language research articles in literature, linguistics and history have pointed out the contrast between the relative lack of clear structural conventions in these fields compared to the much more standardized structures in medicine and the natural sciences [33,53]. Jonas Bakken's study of articles from 1937 to 1957 in Edda, the literature journal included in our corpus, even points to the lack of any form of textual norms for introduction sections as a feature of note [53] (p. 287). Thus, although the research in this area is limited, it is possible to suggest that variety or the lack of a clear template, might in itself be a characteristic quality of traditional introduction sections in the SSH.

It is important, though, to keep in mind that the same might be said of Anglo-American texts in the SSH, given that the CARS model represents a particular kind of Anglo-American research writing. Swales and Fredrickson, for example, point out that "the CARS model or its variants, is a construct, perhaps over-influenced by corpora derived from highly-competitive Anglophone science, technology and medicine" [52] (p. 18). This suggestion highlights that the model is indeed culturally and disciplinary specific and perhaps not particularly well-suited to fully capture introductions in

research articles from other disciplines and contexts. Yet, we found the model useful for purposes since our goal was not to look for whether the introductions adhered to the model per se but to make comparison of specific features of texts across time possible. We could of course have used other features for analysis but due to the widespread use of the CARS model, it offered us rich material for comparison and interpretation.

*2.3. Analysis*

We manually examined all articles in our corpus. We included only research articles and excluded editorials, book reviews, commentaries and other types of genres that the journals themselves did not specifically label "research article" [forskningsartikkel]. We included both theoretical and empirical research articles. This process left us with a corpus of 136 articles. We then identified the introduction sections of each article. Many of the articles had introduction sections that were explicitly set off by a heading called "Introduction." Other articles did not have such a heading but started with a section without a heading to be followed by a section with a heading, making it clear that the first section served an introductory purpose. In five cases, the articles included no clear section breaks whatsoever. In these cases, members of the research team read the articles individually and then conferred with each other to identify the segment of the text that seemed to serve an introductory purpose.

Each introduction section was analysed using a fixed set of questions. The questions included: title of article, the nationality of the institutional affiliation of the author, language of the article, number of paragraphs in the introduction section, whether the article used subheadings or not, which moves from the CARS model were present in each paragraph and the lexical realizations of the various moves.

The authors, who both have Norwegian as their first language, analysed the introduction sections independently and we stored our results in a Google form document. Once we completed our analyses individually, we compared our results. In the cases in which our analyses diverged, we analysed the texts again and discussed our analyses until we reached consensus. We also each looked for lexical realizations of the different moves and compared these, discussing our analyses of these until we reached consensus.

In our analysis, we used the moves from the CARS model as a way to look systematically for rhetorical patterns. Our analysis focused on moves and move cycles rather than on the individual steps of the moves. We made this choice because our goal was to look for structural changes at a more global level. More specifically, we looked for move cycle patterns. Such patterns emerge from the way moves are ordered and any cycles and repetition of moves. For example, as the numbers included in the CARS model indicate, the expected order of moves is M1-M2-M3. As Swales has pointed out, authors order and repeat moves in a great variety of ways [37]. We thus used variation of move cycle patterns as a way to look for degrees of variety in our material. In this way, the recurrence of a specific order of moves or cycles of moves would be an indication of typical structures. In our analysis, then, we started by identifying occurrences of the various moves and noting in what order they appeared.

In order to show how we applied the analytical instrument to the texts in our corpus, we provide one example of our analysis of one introduction below. There are a range of ways of realizing the different moves and numerous ways of ordering them. This example shows only a few of the possible ways to realize the various moves but the point of this example is to show a concrete example of our analytical process. Our example is the article "I skyggen av kanon. Empiri som utfordring i feministisk litteraturvitenskap" [In the shadow of canon. The empirical as challenge in feminist literary scholarship] by Anne Birgitte Rønning from Edda, 2014.

The introduction consists of four paragraphs and Rønning opens her article with a quote by Toril Moi, a well-known feminist literary theorist, from 1979. In the quote, Moi claims that the "main problem" with feminist literary scholarship is a lack of discussion about theory and methods. Rønning uses this quote as indicative of a larger trend of understanding the field as undertheorized.

The author next reviews the scholarship since 1979 and then and concludes in the middle of the second paragraph:

> *Som all nyere litteraturvitenskap bærer også den feministiske litteraturvitenskapen preg av stadig skiftende teoretiske perspektiver, og både appellen om mer teori og større teoribevissthet, og debatter om og avklaringer av utfordringene i feministisk litteraturteori har jevnlig gitt gjenlyd. Ikke minst har både Toril Moi og Ellen Mortensen levert viktige teoribidrag til feministisk forskning (bl.a. Moi 1985, 1987, 2008; Mortensen 1996, 2001, 2002; Mortensen m.fl. 2008; se også Langås 2001; Iversen 2002; Jegerstedt 2008a). Slik er dagens forskningsfelt både konsolidert og beriket….*
>
> *[As all newer literary scholarship, feminist literary scholarship is also characterized by ever shifting theoretical perspectives. The call for more theory and greater theoretical awareness, as well as discussions and clarifications regarding the challenges of feminist literary scholarship have regularly resonated in the field. Toril Moi and Ellen Mortensen, have in particular provided important theoretical contributions to feminist research (see for example Moi 1985, 1987, 2008; Mortensen 1996, 2001, 2002; Mortensen et al. 2008; see also Langås 2001; Iversen 2002; Jegerstedt 2008a). In this way, the research field is currently consolidated and enriched…]*
>
> EDDA-2014–15, p.279

This is an example of a typical Move 1 where the author "establishes a research territory" by reviewing, characterizing or describing existing research on the topic of the article. Rønning describes the field as "consolidated," that is, she suggests that this is a common way of understanding what the field looks like.

Then, in the middle of the second paragraph, Rønning begins to explain that the focus on theory has left other potential areas for investigation unexplored. Specifically, Rønning says that discussions about what appropriate empirical matter for feminist literary scholarship should be have been absent. By the end of paragraph 3, Rønning points out a perspective that she sees as missing from the field:

> *Uavhengig av skiftende teorier har den litterære teksten og analysen av den vært det stabile litteraturvitenskapelige sentralpunktet. (...) Når vi skal reflektere over status og utfordringer for dagens feministiske litteraturvitenskap, kan det derfor være nyttig å sette søkelyset nettopp her—i det mest selvsagte, minst debatterte hjørnet i det vitenskapelige trekantforholdet teori–metode– empiri. Hva er egentlig forskningsfeltets objekter? Hva vil vi med materialet, og hvordan skal vi håndtere det?*
>
> *[Regardless of shifting theories, the literary text and textual analysis have remained firmly at the centre literary scholarship. (…) When we reflect on the status and challenges of contemporary feminist literary scholarship, it may be useful to focus precisely on this—on the most obvious, least discussed corner of the triangulated relationship between theory-method-empirical material. What are the objects of the research field really? What do we want with this material and how should we handle it?]*
>
> EDDA-2014–15, p. 279

In this excerpt, the author points out something that she thinks existing research has not adequately addressed. By pointing out a feature that is at the same time the "most obvious" but "least discussed," the author goes on to list questions that the she considers neglected. In so doing, the author "establishes a niche" by defining and pointing out an area that is in need of further investigation.

Immediately following the passage quoted above, Rønning describes how her article will address this under-discussed area. To speak in terms of the CARS model, she "occupies the niche" by stating that the purpose of the article is to focus on what she has identified as lacking: the empirical material in feminist literary scholarship. The fourth paragraph of the introduction starts out in this manner:

*Med utgangspunkt i egne erfaringer, bl.a. fra tre kvinnelitteraturhistoriske prosjekter, vil jeg i det følgende argumentere for at empirien alltid har vært en utfordring for feministisk litteraturvitenskap, og at utfordringen aktualiseres med utviklingen av digital humaniora.*

*[*Using my own experiences as a starting point, among other things, I have been involved in three projects about women's literary history, I will in the following argue that the empirical material has always been a challenge for feminist literary scholarship and that the development of digital humanities has made this challenge more pressing.*]*

EDDA-2014–15, p. 279

This excerpt is followed by a preview of the section that follows the introduction and this concludes the introduction section. This introduction is, then, an example of an introduction that follows the structure of the CARS model in the expected order: Move 1, Move 2 and Move 3. As we make clear in the results section, this structure was not common in our material but this introduction allows us to show examples of the moves in the model.

In the rest of the text, we refer to the moves in the model by using descriptive statements rather than Move 1, Move 2 and Move 3 to make it easier to read for readers who might not be intimately familiar with the model. We have had to make certain adjustments to Swales's descriptors to make it fit better with our corpus. Instead of using "establishing a research territory" about Move 1, we use "presentation of background." This is because in quite a few cases, the work of Move 1 is realized by describing a general background or context for the topic without referring to existing research. For Move 2, we use Swales's term "establishing a niche" because this description captures the function of this move in our corpus as well. For Move 3, we use the term "explicit statement of aim or purpose," rather than "occupying the niche." This is because our corpus includes many examples of introductions that do not establish a niche. In these cases, the phrase "occupying the niche" does not adequately capture the function of such moves since there is no niche to occupy.

## 3. Results

This section highlights our findings in terms of the three dimensions that emerged as central to our research question: what changes, if any, could be found in the rhetorical structure of the introduction sections from 1994 to 2014? The dimensions that appeared most clearly in our analysis were: (1) an increase introductions that establish a niche, (2) the continued and even increased, variation of overall rhetorical organization and (3) a decrease of introductions without explicit statements of the article's overall aim or purpose.

### 3.1. The Number of Introductions That Establishes a Niche Has Increased in All Fields from 1994 to 2014

The clearest change in our material is an increase in the number of introductions that establish a niche in relation to previous research. As Table 2 shows, in the 1994 corpus 16 (23%) of the introductions included this move, while 30 (45%) of the introductions in the 2014 corpus included this feature. It is important to note that our material shows that a majority of articles both in both years do not include passages that establish a niche. Yet, our material also suggests a clear tendency that the inclusion of this move has become more common in all the journals included in this study.

**Table 2.** Number and percentages of introductions that establish a niche.

| Journal title | Discipline | Introductions That Establish a Niche, No., (%) | |
| --- | --- | --- | --- |
| | | 1994 | 2014 |
| NPT | Education | 4 (11) | 11 (31) |
| SID | Sociology | 4 (27) | 8 (57) |
| EDDA | Literature | 7 (26) | 11 (65) |
| Total | | 16 (23) | 30 (45) |

This trend becomes even more pronounced when we include versions of niche moves that do not fully fit with Swales's model. The CARS model indicates various ways that a niche may be created and emphasizes the need to make space in a particular field. This may be done in various degrees of strength, moving from question raising, to indicating a gap, to direct critique of previous research. In our material, we found several instances where the niche is established, not in a field of research but within a particular set of materials. For example, a tension within a novel under examination or a particular policy document. Rather than pointing out something that is unexplored or problematic in previous research, this move points to something unexplored in the object of analysis itself.

Here are two examples that show the difference in these two versions. First an example that shows a standard version of establishing a niche:

*Lobbyismens betydning i norsk politikk har generelt fått liten forskningsmessig oppmerksomhet (Espeli 1999; Pettersen 2009; Haug 2010). Dette gjelder ikke i mindre grad for kulturpolitikken. I denne artikkelen skal jeg undersøke hva lobbyisme innebærer på det kulturpolitiske området.*

*[In general, there has been little research interest in the influence of lobbyism in Norwegian politics (Espeli 1999; Pettersen 2009; Haug 2010). This is even more true when it comes to cultural policy. In this article I will investigate the role of lobbyism in the field of cultural policy.]*

SID-2014–3, p.67

Here, there is an explicit evaluation of previous research in a very typical manner. By stating that "there has been little research interest in," the author points out a shortcoming in the research field and then proceeds to describe how the article will address this shortcoming. In the following example, however, the author points to a tension in the object of analysis:

*Og ingen av dei [andre mannlige lyrikere] går nærmare inn på det som synest så altoppslukande hos dei kvinnelege lyrikarane, nemleg den påfallande opplevinga av den eige kroppen som levande, som fylde i verda.*

*[And none of them (other male poets) hone in on what seems so all-consuming for the female poets,* i.e., *the striking experience of the body as alive, as a presence in the world.]*

EDDA-2014–12, p. 211

Here, there is no reference to a gap in research or in the research community. Rather, there is an observation that no male poets other than Tor Ulven, (the poet who is the subject of the article) writes about the typical female theme of bodily experiences. In other words, there is an unexplored area in the object of study but there is no explicit mention of how previous scholars have approached this area.

As Fredrickson and Swales [52] note in their analysis of introduction sections in Swedish linguistics, there is an opportunity to develop a niche in the research territory by saying that this has not been noted by other scholars before or the like but the author has chosen not to so. In our material, such passages appeared in enough cases and across fields (6 instances in the 1994 corpus and 10 instances in the 2014 corpus) that we think it is worth pointing out as a distinct feature that we have categorized as an "implicit niche." As shown in Table 3, if we count these instances, 22 introductions (31%) established a niche in the 1994 corpus and 40 introductions (60%) included this move in the 2014 corpus.

**Table 3.** Number of introductions that establish a niche, including implicit niche variants.

| Journal Title | Discipline | Introductions That Establish a Niche, No., (%) | | Including Implicit Niche Variants No., (%) | |
|---|---|---|---|---|---|
| | | 1994 | 2014 | 1994 | 2014 |
| NPT | Education | 4 (11) | 11 (31) | 6 (21) | 15 (43) |
| SID | Sociology | 4 (27) | 8 (57) | 6 (40) | 10 (71) |
| EDDA | Literature | 7 (26) | 11 (65) | 9 (33) | 14 (82) |

| Total | 16 (23) | 30 (45) | 22 (31) | 40 (60) |

*3.2. Variation in Overall Rhetorical Organization Contiues*

The other most prominent finding in our material is that there is a great variation in overall rhetorical organization in both the 1994 corpus and the 2014 corpus. Before we describe this variety, it should be noted that we only had five occurrences in our entire corpus that were organized in ways that we were not able to analyse by using the model and hence classified as "other." The fact that we could apply the CARS model to such a large portion of our corpus could be seen as an indication that a high degree of standardization is in fact in place. One example of an introduction that we classified as "other" in our material was from the sociology journal from 1994, the article Byvisjoner og byforståelse [Visions and Perceptions of the City] by Per Morten Schiefloe. The article discusses various historical understandings of the city and urban life in a Scandinavian context. Identifying any moves from the CARS model was difficult because of the absence of metadiscursive elements that signal introductory functions. There is a subheading labelled "1. Visjoner og utopier" [1. Visions and utopias] but this section stretches over several pages and takes the form of a discussion of different understandings and perceptions of the city. In this way it is difficult to see how it serves as a prefacing mechanism. Instead, the section seems to jump right into the material without distinguishing a separate "background" into which the subject matter is placed. It is, perhaps, this lack of hierarchical positioning in terms of what is general and what is specific that makes the CARS model difficult to apply because it is hard to locate a bounded "territory" or to distinguish between "background" and "foreground." The other cases that we classified as "other" had a range of features that made classification according to the CARS model difficult but a common denominator is this lack of indicators of hierarchical movement, between specific and general; background and foreground. Instead, they seem characterized by points that are organized horizontally rather than hierarchically.

Since this was only a feature of five introductions, this remains a speculative hypothesis at this point but it is interesting that our 2014 corpus contain no examples of introductions that we found difficult to categorize using the CARS model, thus perhaps tempering our overall conclusion about variation. At the very least, one could say that the kind of variety we are able to find using the CARS model as our instrument, is variety within quite a narrow scope.

Even so, an analysis that focused on move cycle patterns gave us the ability to look for the emergence or disappearance of any prominent patterns. For example, the "ideal" structure as presented in the CARS model, M1-M2-M3, is infrequent in both the 1994 corpus and the 2014 corpus. As Table 4 shows, 5 (7%) of the introductions in the 1994 corpus followed this pattern, while 11 (17%) of the introductions in the 2014 corpus used this pattern.

**Table 4.** Number and percentages of introductions following an M1-M2-M3 pattern.

| Journal | Discipline | Introduction Sections, No. | | | M1-M2-M3 Cycle Pattern, No., (%) | |
|---------|-----------|------|------|-------|------|------|
| | | 1994 | 2014 | Total | 1994 | 2014 |
| NPT | Education | 28 | 35 | 63 | 2 (7) | 8 (23) |
| SID | Sociology | 15 | 14 | 29 | 3 (20) | 2 (15) |
| EDDA | Literature | 27 | 17 | 44 | 0 (0) | 1 (6) |
| Total | | 70 | 66 | 136 | 5 (7) | 11 (17) |

In other words, while there is an increase from 1994 to 2014, our material does not indicate that this move cycle pattern holds a particularly prominent position in either year.

The most frequently used rhetorical organization is one in which the author presents some kind of background and then includes a statement of purpose or aim, without establishing a niche. In the CARS model, this pattern is indicated by an M1-M3 sequence. This pattern appears 43 times (32%) in our corpus as a whole. The frequency of this move cycle pattern has decreased from 24 instances (39%) in the 1994 corpus to 19 instances (29%) in the 2014 corpus.

While this type of pattern is the one that appears most frequently, the most typical feature in our material is variation. That is, most introductions use a rhetorical organization that few, if any, other

articles use. In 1994, 28 instances (40%) of the introductions in our material consisted of move cycle patterns that appeared only one or two times in a given journal in a given volume. In 2014, the percentage of introductions that employed such infrequently used move cycle patterns had increased to 37 instances (56%). Especially the 2014 volume of Edda shows great variation. Of the 17 articles in the volume, only three types of move sequences appeared twice (1-3-1-3, 1-3-2 and 1-3). The rest of the introductions show patterns that appear only once. In other words, a main finding in our material is that variety in rhetorical organization is typical in both the 1994 corpus and the 2014 corpus. Again, it is worth noting that we did not find any occurrence of introductions that we classified as "other" in 2014, while we found five in 1994. This might suggest that introductions that do something beyond the moves of the CARS model have become less common but our material is too small to tell if this is a fluke or a trend. We have included a full overview of the different move cycles in our corpus in Appendix B.

*3.3. Decrease of Introductions without an Explicit Statements of the Article's Overall Aim or Purpose*

The third prominent finding in our analysis is that the number of introductions without an explicit statement of purpose has decreased. These introductions typically only present some kind of background information about the topic. NPT-1994-3, p. 114 offers an example of this kind of introduction which is widely used in the 1994 volume of the education journal. The introduction has no headline and follows an abstract in italics. The article's title is "Lærerrollen og religion i skolen— et forsøk på anvende Girards teorier på religionsdidaktikken" ["The teacher role and religion in school—an attempt at applying Girard's theories on teaching religion"] and is written by Per Bjørnar Grande. The introduction consists of three short paragraphs giving background on Girard and his theory of mimetic desire and scapegoat mechanism. While the third and final paragraph sums up the central work by Girard used in the article *Deciet, Desire & the Novel*, there is no attempt at connecting the subject of the article as stated in the title to the information given in the introduction and no explicit statement of the article's purpose. This kind of introduction might draw on a tradition in which the conclusion should be arrived at in the conclusion rather than stated early on in the text.

In the corpus as a whole, this background-only structure appears in 25 instances (18%). The occurrence of this organization decreases from 19 instances (27%) in 1994 to only five instances (8%) in 2014. As Table 5 shows, the decrease of this pattern is particularly pronounced in the education journal where it appears in 12 instances (43%) in the 1994 volume and drops to four instances (11%) in the 2014 volume. In the 1994 volume of Edda, six of the introductions (22%) have a background-only pattern. By 2014, only one instance (6%) of this type of introduction can be found.

**Table 5.** Introduction sections without an explicit statement of aim or purpose.

| Journal | Discipline | Introduction Sections, No. | | | Without Explicit Aim, No., (%) | |
|---------|-----------|------|------|-------|---------|---------|
| | | **1994** | **2014** | **Total** | **1994** | **2014** |
| NPT | Education | 28 | 35 | 63 | 12 (43) | 4 (11) |
| SID | Sociology | 15 | 14 | 29 | 1 (7) | 0 (0) |
| EDDA | Literature | 27 | 17 | 44 | 6 (22) | 1 (6) |
| Total | | 70 | 66 | 136 | 19 (27) | 5 (8) |

**4. Discussion**

In this study, we wanted to look for possible changes in writing conventions in another language than English during a period of multilingual publication patterns. We examined the rhetorical structures of research article introductions published in three Norwegian-language journals, using the CARS model as our instrument of analysis. Our corpus consisted of 70 introduction sections from 1994 and 66 introduction sections from 2014 from the fields of education, sociology and literature. Our findings highlight three main trends: an increase of introductions that establish a niche within existing research, continued variation in overall rhetorical structures and a decrease of introductions without an explicit statement of the article's aim or purpose.

Our discussion starts with interpreting our findings in relation to relevant studies of research article introductions as outlined in Section 2.2. and then moves on to position our findings in relation to previous work on the role of language, discipline and culture in shifting writing conventions as outlined in Section 1.2.

In previous studies of introduction sections, the relative absence and/or presence of passages that establish a niche has received much attention in cross-cultural analyses of introduction sections. Many of these studies have noted the relative absence of such passages in non-English texts [42,48,50–52,54]. The proposed reasons for this include reluctance to criticize other scholars due to solidarity with the research community [45,48], cultural mindsets [50,51] and the size of the national discourse community [52].The latter point builds on the ecological metaphor of the CARS model to suggest that in research communities that are relatively small, there is less competition for research space and hence less need to create this space rhetorically [52].

Other studies have found that passages that establish a niche are common also in other languages than English. Both Sheldon [46] and Wannaruk & Amnuai [44] found that such moves have become a standard feature in applied linguistics journals published in Spanish and Thai, respectively. Sheldon suggests two explanations in the Spanish case. One reason is the increased influence of English-language rhetorical conventions on Spanish rhetorical conventions. The other reason, Sheldon says, is that applied linguistics in Spain has become a more crowded field, increasing the competition for research space [46].

Our analysis found that passages establishing a niche are quite commonly missing both in the 1994 corpus and in the 2014 corpus. However, the relative absence or presence of such statements varies from field to field. In the education journal, for example, there are far more instances of introductions that establish a niche in the 2014 corpus than in the 1994 corpus, yet the majority of articles in the 2014 corpus, 57%, still do not contain such statements. In other words, our analysis suggests that in education, including passages that establish a niche has become more common but it is not yet the norm. In literary studies, however, there has also been an increase. By far the majority of the articles, 82%, in the 2014 corpus do include such passages. As we will discuss in more detail below, we are hesitant to attribute these findings to any one particular aspect, rather, like Sheldon, we expect a combination of factors, including, disciplinary developments to be involved.

The increase of introductions that establish a niche combined with the decrease of introductions that do not contain an explicit statement of purpose suggests that more explicit positioning has become a rhetorical requirement. This trend is in line with previous diachronic studies, such as Shaw and Vassileva's study of journal articles in the field of economics in four different languages: Danish, Bulgarian, English and German [22]. Shaw and Vassileva examine a much wider array of textual features than our study does. Explicit positioning in terms of indicating a niche and stating an explicit purpose was only one of the aspects they investigated. In their corpus, the proportion of articles across languages indicating a niche and stating an explicit purpose moved from being common in 53% of the journals in 1900 to becoming a standard feature in 95% of the articles by early 2002 [22] (p. 298). Because these shifts have happened at similar times across languages, they argue that these changes might primarily be understood as a result of greater "specialization" and "professionalization" of the discipline of economics [22] (p. 300). However, they also suggest that the growing status of English as a language of research and the decline of the status of German and Russian might have influenced some of the changes they observed in the texts written in Danish and Bulgarian.

Our study also indicates that explicit positioning seems to vary with disciplines, suggesting that the field of education has gone through a greater change than sociology or literary studies. This might be an indication of larger epistemic changes within education, a field which has become considerably more research-intensive over the last few decades.

Salager-Meyer et al. [28], who examined textual and rhetorical features of expressions of "academic conflict" in Spanish, French and English medical discourse across the 20th century, also propose that the changes they observed can be attributed to the combination of disciplinary developments and the rise of English as the dominant language of research. They did not specifically

focus on introduction sections or move structures, yet their conceptualization of academic conflict as a type of positioning in the research community makes their study relevant for our results. In many instances, establishing a niche is similar to what they identify as "academic conflict," for example in pointing out problems with previous research or proposing counter arguments. They note a shift in Spanish discourse from the 1990s in which an earlier "vigorous, often times passionate, acerbic, author centered and frequently scathingly sarcastic tone" (p. 235) has been superseded by "a rather gentle, neutral, dispassionate, matter-of-fact and apparently indifferent tone" (p. 234). They suggest that these changes are due the influence of the dominance of English-language research and the appearance of writing handbooks and writing courses based on Anglo-American writing traditions [28] (p. 240).

Bennett's examination of Portuguese history writing from 1998 to 2013 documents a similar shift "from French or Romance discursive models to towards English ones" [26] (p. 34). Like Salager Meyer at al., she attributes these changes to the dominance of English as the language of research: "The requirement to produce texts in English obliges authors to develop different mental habits (different lexical categories; different ways of organizing material at the grammatical and textual levels) and it is natural that this should eventually filter through to their mother tongue writing" [26] (p. 36). Both Salager-Meyer and Bennett, then, firmly conclude that Portuguese and Spanish academic discourse is changing because these discourses are adopting Anglo-American models.

Our material does not allow us to draw such clear conclusions. One reason might be that the more drastic changes observed by these studies is due to the fact that Spanish and Portuguese writing traditions have been further from the Anglo-American traditions than Norwegian traditions have been in the first place. However, the fact that our analysis offers less clear conclusions might also be explained by the limitations of our study.

One obvious limitation is the narrow focus on the move structure in introduction sections. Bennett, Salager-Meyer et al. and Shaw and Vassileva examine a whole range of textual features and they have thus been able to grasp a richer sense of the texts. A study looking at a wider set of features such as the use of pronouns, reference patterns and sentence structures would have given us a more in-depth and holistic sense of textual and rhetorical developments to yield a firmer basis for conclusions. Also, since we have not compared our material to English-language journals in the same fields as those included in our corpus, in the way that both Salager-Meyer et al. and Shaw and Vassileva do, we cannot say whether the trends we have identified are specific to Norwegian or whether they may be found in English academic discourse in these fields as well. In sum, our study is quite limited in scope and range and this means our conclusions must be more cautious as well.

Importantly, our corpus cannot in and of itself answer why the trends we have identified are happening. Although, we have emphasized the historical and cultural context of our corpus, we cannot draw any causal links to show that this context caused the trends we have observed. Drawing on previous studies, however, we find it reasonable to suggest that disciplinary developments, research-funding policies, as well as English-language dominance all play a part in explaining the trends we have identified. However, in order to get a more precise understanding of why such writing conventions change, our analysis of texts needs to be followed up by studies that focus on writing practices. We thus think that there is reason to look more closely into what Bennett describes as a "natural" development of "different mental habits" among multilingual writers exposed to Anglo-American research writing. Such studies would necessarily have to use other methods than textual analysis, such as for example historical and ethnographic studies of academic disciplines, academic journals, review and editorial practices and scholars' writing and research practices.

For all of our study's limitations, however, we do think that examining texts across time is an important starting point for more in-depth qualitative studies of the various institutional and individual practices that shape these changes.

## 5. Conclusions

We started this article by claiming that research on texts and practices of multilingual scholars should include studies of how these scholars work in other languages than English to avoid an

English-centric conceptualization of multilingualism. As we note above, this particular study is but a small step towards such a focus. The limited scope of our analysis and empirical material do not lend themselves to generalizations. Our findings also indicate fewer sweeping changes than those that have been taking place in Romance academic discourses, where the term "epistemicide" appears to capture the demise of traditionally Romance discursive ideals. Yet, we believe that it is important to understand the changes we did find as part of a complex dynamic between languages and discursive traditions and ideals. As Bennett and several others have argued, there are political and ideological reasons for being cautious of an uncritical adoption of Anglo-American models if they are understood as "better writing" than the models they are replacing. From such a perspective, other forms of writing and thinking are positioned as "deficient" rather than as indicative of alternative ways of constructing and representing knowledge.

We thus need more studies that attend to the dynamic between the different languages and discursive traditions that multilingual scholars work in. This is particularly important for social sciences and humanities, where publishing in more than one language is the norm rather than the exception. In such work, it might be necessary to put scholarship about various non-English languages (which perhaps has been less available in English-language journals) in conversation with scholarship with a focus on English as additional language. As of today, those literatures do not often speak to one another, even though their areas of inquiry certainly overlap.

To overcome one of the limitations of our study, such research should not focus on texts only. Rather, we need studies that attend to both texts and writing practices in order to understand the institutional structures, policies and writing practices that produce the texts. Thus, we join scholars who call for studies that examine precisely cross-linguistic textual practices [3,13–16].

For practitioners in fields such as English for Research Publication Purposes, we echo Carmen Perez-Llantada's argument for the importance of a "multiliterate rhetorical consciousness-raising pedagogy" [55] (p. 15). Through such consciousness-raising, we might be able to make better use of multilingual scholars' linguistic resources. In this perspective, multilingual scholars who use English as an additional language might consider the expertise they have in a different language than English a resource that allows for flexibility, greater rhetorical awareness and agency rather than as an impediment that must be "overcome." In order to design and implement such a pedagogy, collaborations between colleagues who work with academic discourse in different languages should be encouraged and welcomed.

**Author Contributions:** K.S. and I.L.Ø. jointly initiated the topic, designed the study, analysed the material, wrote and revised the manuscript.

**Funding:** This research received no external funding.

**Acknowledgments:** The authors would like to thank colleagues at OsloMet for reading and commenting on drafts of this article.

**Conflicts of Interest:** The authors declare no conflict of interest.

## Appendix A

Corpus examined for "Multilingual Research Writing beyond English: The Case of Norwegian Academic Discourse in an Era of Multilingual Publication Practices."

**Table A1.** Norsk pedagogisk tidsskrift 1994.

| | Author | Title |
|---|---|---|
| **NPT-1994-1** | Lars Monsen | Ungdomstid og skoletid |
| **NPT-1994-2** | Brit Ulstrup Engelsen | Lærlingeordningen—et kritisk punkt i Reform-94. |
| **NPT-1994-3** | Kåre Heggen | Ungdom og endra kvalifisering |
| **NPT-1994-4** | Lidveig Bøe | Pengane eller vitet? Skuleungdoms deltidsarbeid sett i lys av utdanningspolitiske utviklingstrekk. |
| **NPT-1994-5** | Gunnar Bergendal | Lärarens kallelse |
| **NPT-1994-6** | Arild Gulbrandsen | Paradigmeskifte i lærerutdanningen? |

| | | |
|---|---|---|
| **NPT-1994-7** | Truls Kobberstad | Problemløsning i norsk skolematematikk: Den Baconske drøm om en oppdagelsens algoritme? |
| **NPT-1994-8** | Torlaug Løkensgard Hoel | Språk i læringsprosess og klasserom |
| **NPT-1994-9** | Ole Andreas Isager | Biologifaget i grunnskolens fagplaner. Den historiske utviklingen i et vitenskapsteoretisk perspektiv. |
| **NPT-1994-10** | Per Bjørnar Grande | Lærerrollen og religion i skolen—et forsøk på å anvende Girards teorier på religionsdidaktikken. |
| **NPT-1994-11** | Yngvar Ommundsen | Helsefremmende arbeid i skole/kroppsøving—et pedagogisk sosiologisk perspektiv |
| **NPT-1994-12** | Harald Lauglo | Fagvalg i ungdomsskolen. Utviklingstrekk i elevenes valg. |
| **NPT-1994-13** | Richard Haugen | Trivsel, selvoppfatning og sosialt miljø i klassen. En analyse av sammenhenger. |
| **NPT-1994-14** | | Fravær i den videregående skole. |
| **NPT-1994-15** | Tore Gunnar Sandve & Torunn Bjørkmo | "De uforklarlige underyterne" og en elevtilpasset skole |
| **NPT-1994-16** | | Høstbarn i norsk skole. Alder ved skolestart og risiko for skolevansker. |
| **NPT-1994-17** | Stig Flaata | Strategi og pedagogisk handling. Spillteorien og dens anvedendelse på det pedagogiske området. |
| **NPT-1994-18** | Kjetil Steinsholt | Retten til å være annerledes. Del 1. Reflkesjoner over M. Foucaults liv og virke. |
| **NPT-1994-19** | Halvor Bjørnsrud | Fra høringsutkast (H92) til ny læreplan (L93) |
| **NPT-1994-20** | Britt Ulstrup Engelsen | Ny læreplan for skoleverket |
| **NPT-1994-21** | Synnøve Skjong | "Trygg i tradisjonen og budd til bridge?" Ei semiotisk lesing av Læreplan for grunnskole og vidaregåande opplæring, generell del. |
| **NPT-1994-22** | Roald Nygård | Forskningskvalitet: Lang mer enn et spørsmål om metode. |
| **NPT-1994-23** | Berit Bae | "Hei løve, er du farlig eller grei?" Om lekende samspill |
| **NPT-1994-24** | Kjetil Steinsholt | Retten til å være annerledes. Del 2. |
| **NPT-1994-25** | Knut Imerslund | Jens Bjørneboe og pedagogikken. |
| **NPT-1994-26** | Vegard Nore | Migrasjonspedagogikken som miskommunikasjon. Et symptom på avsporing av rasismedebatten. |
| **NPT-1994-27** | Thor Ola Engen | "Reflection-in-action" eller "action in reflection"? Vygotskij utypder Wittgenstein. |
| **NPT-1994-28** | Liv Duesund | Måling av selvoppfatning hos barn |

**Table A2.** Norsk pedagogisk tidsskrift 2014.

| | Author | Title |
|---|---|---|
| **NPT-2014-1** | Kåre Heggen and Finn Daniel Raaen | Koherens i lærarutdanninga |
| **NPT-2014-2** | Kirsti Vindal Halvorsen | Utvikling av partnerskap i en femårig lektorutdanning—sett fra et økologisk perspektiv |
| **NPT-2014-3** | Kirsten Sivertsen Worum | Veiledning, kunnskapssyn og danning |
| **NPT-2014-4** | Ruth Ingrid Skoglund | Danning i barnehagen: Hva kan danningens «mer enn» være? |
| **NPT-2014-5** | Øistein Anmarkrud, Ivar Bråten and Helge I. Strømsø | Strategisk kildevurdering av multiple tekster: Utbytterikt, men krevende |
| **NPT-2014-6** | Britt Oda Fosse og Sylvi Stenersen Hovdenak | Lærerutdanning og lærerprofesjonalitet i spenningsfeltet mellom ulike kunnskapsformer |
| **NPT-2014-7** | Tore Witsø Rafoss og Hilde Witsø | Fagenes krav og lovens bokstav En kvantitativ undersøkelse av prøvenemndene på Agder |
| **NPT-2014-8** | Evelyn Eriksen | Prinsippet om barnets beste i barnehagen |
| **NPT-2014-9** | Roald Iversen | Utdanning og ulikhet i Norge Opprettholder vi en tradisjonell lagdelingsstruktur, eller beveger vi oss mot en meritokratisk klassestruktur? Et historisk tilbakeblikk |
| **NPT-2014-10** | Bente Vatne and Liv Gjems | Barnehagelæreres arbeid med barns språklæring |
| **NPT-2014-11** | Janne Fauskanger and Reidar Mosvold | Innholdsanalysens muligheter i utdanningsforskning |
| **NPT-2014-12** | Finn Skårderud and Liv Duesund | Mentalisering og uro |
| **NPT-2014-13** | Sigrun K. Ertesvåg | Profesjonelle kulturar og uro i skulen |
| **NPT-2014-14** | Marit Øien Sæverud | Lærarar sine erfaringar med aksjonslæring som metode for å utvikle læringsleiing |
| **NPT-2014-15** | Mari-Anne Sørlie and Terje Ogden | Mindre problematferd i grunnskolen? Lærervurderinger i et 10-års perspektiv |
| **NPT-2014-16** | Magnar Ødegård | Uro i skolen og den menneskelige væremåte |

| | | |
|---|---|---|
| **NPT-2014-17** | Tone Skinningsrud | Struktur og prosess i norsk utdanning på 1990-og 2000-tallet—Et makrososiologisk perspektiv |
| **NPT-2014-18** | Kåre S. Fuglseth | Religion og pedagogikk—Eit postsekulært syn på religion i skulen |
| **NPT-2014-19** | Tone Sævi | Eksistensiell refleksjon og moralsk nøling—Pedagogikk som relasjon, fortolkning og språk |
| **NPT-2014-20** | Olav Hovdelien and Gunnar Neegaard | Gudstjenester i skoletiden—rektorenes dilemma |
| **NPT-2014-21** | Ida Marie Høeg og Hans Stifoss-Hanssen | «Nå er du hos Gud: Jeg vet ikke hvilken Gud du er hos, men du har det sikkert veldig bra |
| **NPT-2014-22** | Kirsten Grønlien Zetterqvist og Geir Skeie | Religion i skolen; her, der og hvor-som-helst? |
| **NPT-2014-23** | Bengt-Ove Andreassen | Religionslæreren—en rolle i endring |
| **NPT-2014-24** | Ann Midttun | Biter og deler av islam |
| **NPT-2014-25** | Geir Winje | Elevers lesing av bilder i RLE |
| **NPT-2014-26** | Inger Margrethe Tallaksen and Hans Hodne | Hvilken betydning har læremidler i RLE-faget? |
| **NPT-2014-27** | Dag Husebø | Tro-og livssynsfag i Skandinavia—en sammenligning |
| **NPT-2014-28** | Robert Jackson | «Veiviseren»: En presentasjon av Europarådets retningslinjer for religions-og livssynsundervisning—(«Signposts»: Dissemination of Council of Europe Policy on Education about Religions and Non-religious Convictions) Oversatt til norsk av Marie von der Lippe |
| **NPT-2014-29** | Sylvi Stenersen Hovdenak | Utdanningspolitikk, forskning og kunnskapsformer—Globale og nasjonale tilnærminger |
| **NPT-2014-30** | Tom Are Trippestad | Visjonærstillingen |
| **NPT-2014-31** | Ove Skarpenes and Ann Christin E. Nilsen | «Making up pupils» |
| **NPT-2014-32** | Kaare Skagen | Digitalisering som statlig avdidaktisering av klasserommet |
| **NPT-2014-33** | Berit Karseth and Jorunn Møller | «Hit eit steg og dit eit steg»—Et institusjonelt blikk på reformarbeid i skolen |
| **NPT-2014-34** | Hilde Wågsås Afdal | Fra politikk til praksis—konstruksjon av læreres profesjonelle kunnskap |
| **NPT-2014-35** | Nina Volckmar | Et blå-blått utdanningspolitisk skifte?—En studie av den utdanningspolitiske retorikken i partiprogrammene til stortingsvalget 2013 og Sundvollen-plattformen |

**Table A3.** Sosiologi i dag 1994.

| | Author | Title |
|---|---|---|
| **SID-1994-1** | Geir O. Rønning | Handling og struktur hos Anthony Giddens |
| **SID-1994-2** | Else Jerdal | Anthony Giddens—kritisk sosiolog eller samfunnsfilosof? |
| **SID-1994-3** | Martin Eide & Karl Knapskog | Samfunnsforskning mellom Cambridge og Dikemark. Struktureringsteoriens posisjon og potensial" |
| **SID-1994-4** | Halvor Fauske | Sosialteori for det neste århundre? En sammenligning av Giddens og Parsons. |
| **SID-1994-5** | Olav Eikeland | Aksjonsforskning—empirisk forskning, organisasjonsutvikling eller filosofi? Om divergenser og konvergenser i et "tverrfaglig felt" |
| **SID-1994-6** | Annick Prieur | Mandighet og biseksualitet i Mexico |
| **SID-1994-7** | Anja Bredal | Annerledes på en annen måte. Kampen om innvandrerskapets kategorier |
| **SID-1994-8** | Per Morten Schiefloe | Byvisjoner og byforståelse |
| **SID-1994-9** | Susan S. Fainstein | Rettferdighet, politikk og utviklingen av urbane rom |
| **SID-1994-10** | Randi Hjorthol | Byen som ramme for kvinners og menns daglige reiser og aktiviteter. |
| **SID-1994-11** | Inger Furuseth | Teorier om sosiale bevegelser. En komparativ analyse. |
| **SID-1994-12** | Craig Calhoun | Hvor nye er "nye sosiale bevegelser"? |
| **SID-1994-13** | Jemima Garcia-Godos | Sosiale bevegelser og utvikling |
| **SID-1994-14** | Ana Devic | Sivile identiteter i nasjonalismens tidsalder. Fredsbevegelser i det tidligere Jugoslavia. |
| **SID-1994-15** | Klaus Rasborg | Henimod en sociologisk teori om postmoderniteten. |

**Table A4.** Sosiologi i dag 2014.

| | Author | Title |
|---|---|---|
| **SID-2014-1** | Anne-Britt Gran | Digitale tider i kulturlivet Om digitalt kulturforbruk, kulturpolitikk og kulturelle etterslep |
| **SID-2014-2** | Roger Blomgren & Jenny Johannisson | Varför regional kulturpolitik? Legitimeringsberättelser i svenska regioner |
| **SID-2014-3** | Sigrid Røyseng | Kulturpolitikk og lobbyisme En case-studie av det dansepolitiske oppsvinget under Kulturløftet |
| **SID-2014-4** | Heidi Stavrum | Hvor mange gullplater henger på veggen? Om danseband og kvalitet |
| **SID-2014-5** | Pål Veiden | Når grenser forsvinner—Europa i det små |
| **SID-2014-6** | Tore Slaatta | Det transnasjonale nyhetsbeitet: En mediesosiologisk utmark i Europa-forskningen |
| **SID-2014-7** | Terje Rasmussen | Politisk offentlighet og legitimitet i EU: Sosiologiens bidrag |
| **SID-2014-8** | Olav Elgvin og Jon Horgen Friberg | Migrasjonssosiologiens svarte boks? Sosialpsykologiske prosesser i møtet mellom innvandrere og det norske samfunnet |
| **SID-2014-9** | Michael Hviid Jacobsen og Jan Brødslev Olsen | Dødens socialpsykologi—perspektiver på døden i samspillet mellem individ og samfund |
| **SID-2014-10** | Tone Schou Wetlesen | Møtesteder og pardannelse |
| **SID-2014-11** | Gunn Imsen og Magnus Rye Ramberg | Fra progressivisme til tradisjonalisme i den norske grunnskolen? Endringer i norske læreres pedagogiske oppfatninger i perioden 2001–2012 |
| **SID-2014-12** | Marie Louise Seeberg, Idunn Seland and Sahra Hassan | "Har vi hatt leksehjelp nå?" Sosial utjevning når alle skal med |
| **SID-2014-13** | Ida Holth Mathiesen, Siri Mordal and Trond Buland | En rådgiverrolle i krysspress? Lokal variasjon og konsekvenser for rådgivningen i skolen |
| **SID-2014-14** | Karin Hellfeldt, Björn Johansson & Odd Lindberg | Mobbning och social stöd från lärare och klasskamrater: En longitudinell studie av barns erfarenheter av mobbning |

**Table A5.** EDDA 1994.

| | Author | Title |
|---|---|---|
| **EDDA-1994-1** | Fritz Paul | Utsynet fra toppen. Tradisjon og forandring i et litterært motiv fra følsomhetens tid til Ibsen. |
| **EDDA-1994-2** | Gunnar Foss | Frå Time til Itaka—Garborg og det greske. |
| **EDDA-1994-3** | Jaqueline Broese van Groencu | Fiksjon og virkelighet. En lesning og av Edith Øbergs Mann i mørke (1939) |
| **EDDA-1994-4** | Rick Lybeck | Three Structural Levels in Johan Falkberget's Christianus Sexus |
| **EDDA-1994-5** | Joel Shatzky& Sedwitz Dumont | "All or Nothing": Idealism in A Doll House |
| **EDDA-1994-6** | Olav Solberg | "Opfostret i historie" Om det historiske i Kristin Lavransdatter |
| **EDDA-1994-7** | Hening Howlid Wærp | Den romantiske hage. Den lesing av Andreas Munchs dikt "Natlig Fart" og Paa Tindsøen" |
| **EDDA-1994-8** | Per Mæleng | Fysiognomier. Kommentarer til kroppen som skiftens scene. Lesning av Knut Hamsuns Sult. |
| **EDDA-1994-9** | Ole Egeberg | Ironiens læsning—læsningens ironi |
| **EDDA-1994-10** | Bjørn Stokseth | Fiksjonsprosa og retorikk. Narrative strategier i Ragnhild Jølsens novelle "Felelaaten i Engen" |
| **EDDA-1994-11** | Poul Houe | Jacob Paludan og Eric Eberlins sceniske utopi. |
| **EDDA-1994-12** | Anna Lyngfelt | Anne Charlotte Edgren Lefflers En räddande engel. En enaktare med drag av 1800-talets tableaux vivants-tradition? |
| **EDDA-1994-13** | Knut Stene-Johansen | Form og tanke i Stepahne Mallarmes Gravdikt |
| **EDDA-1994-14** | Sissel Lie | Medusas stemme |
| **EDDA-1994-15** | Arne Melberg | Hölderlins text |
| **EDDA-1994-16** | Arnbjørn Jakobsen | "Hvad skal jeg ha' at leve for da? Bagefter?" Om Ibsens bruk av bibelallusjoner i samtidsskuespillene |
| **EDDA-1994-17** | Anne Marie Rekdal | Noe skjønt—lokkende—og modig. En lacaniansk analyse av Hedda Gabler. |
| **EDDA-1994-18** | Lisbeth Pettersen Wærp | Dunkelhetens estetikk. En retorisk lesning av Ibsens Når vi døde vågner |
| **EDDA-1994-19** | Anne Heith | Representation och kontext. Skiss till en analys av representation som process i Dag Solstads Roman 1987 |
| **EDDA-1994-20** | Erling Aadland | Episk differens |
| **EDDA-1994-21** | Vigids Ystad | Dikterens syner. ibsen og den moderne sanselighet. |
| **EDDA-1994-22** | Hans Peter Thøgersen | Knut Hamusn og Johannes V. Jensen—det nye mennesket. |

| | | |
|---|---|---|
| **EDDA-1994-23** | Christian Koch | Hvad læser vi for? Er litteraturens interessemoment semantisk eller psykodynamisk? |
| **EDDA-1994-24** | Henning Howlid Wærp | Symbol og allegori hos Paul de Man—Romantikken revisited |
| **EDDA-1994-25** | Peter Tahler | Hvorfor norskamerikansk litteratur? |
| **EDDA-1994-26** | Vasilis Papageorgiou | Willy Kyrklunds Mede från Mbongo: Teorin, skapandet och kosmetiken |
| **EDDA-1994-27** | Anders Pettersson | Om litteraturforskningens objektivitet og relativitet |

**Table A6.** EDDA 2014.

| | **Author** | **Title** |
|---|---|---|
| **EDDA-2014-1** | Beata Agrell | Efter folkhemmet: välfärd, ofärd och samtalets estetik i svensk prosalitteratur under "rekordåren" på 1960-talet |
| **EDDA-2014-2** | Michael Schulte | Kenning, metafor og metonymi Om kenningens kognitive grunnstruktur |
| **EDDA-2014-3** | Giuliano D'Amico | Editore-traditore? Knut Hamsun lest, oversatt og publisert av italienske neofascister |
| **EDDA-2014-4** | Anna Salomonsson | Flugan och förtrycket Det koloniala och patriarkala våldets individuella och universella aspekter i Sofi Oksanens Utrensning |
| **EDDA-2014-5** | Olle Widhe | Det sanna pojkhumöret Krig, lek och trivialisering i Ossian Elgströms pojkböcker |
| **EDDA-2014-6** | Claus Elholm Andersen | Forfatteren og sociologen—om Karl Ove Knausgård og Geir Angell Øygarden |
| **EDDA-2014-7** | Annegret Heitmann | «Til Ostindien eller St. Croix« Cirkulation og kosmopolitisme i 1700-tallets dramatik |
| **EDDA-2014-8** | Unn Falkeid | «Helgeninnen med diktersjelen» Sigrid Undsets Caterina av Siena lest i lys av nyere forskning |
| **EDDA-2014-9** | Mette Mortensen | Mellemværende Om grænseopløsning, tvetydighed og ugennemsigtighed i Søren Kierkegaards »Forførerens Dagbog« |
| **EDDA-2014-10** | Ann Schmiesing | Why Is Hulda Lame? Drama and Disability in Bjørnson's Halte-Hulda |
| **EDDA-2014-11** | Therese Svensson | «Trött på vithet»—intersektionalitet i Dan Anderssons Kolarhistorier |
| **EDDA-2014-12** | Hadle Oftedal Andersen | Auget og den fordømte kroppen Om Tor Ulvens nyansering av Merleau-Pontys kunstfilosofi |
| **EDDA-2014-13** | Pål Bjørby | «En vis Skribent»: F. Poulain de la Barre (1647–1723) og hans tre cartesianske forsvar for kvinnen som hovedkilden til «feminismen» i L. Holbergs dikt «Zille Hans Dotters Gynaicologia» (1722) |
| **EDDA-2014-14** | Jonas Bakken | Disputaser i Edda gjennom 100 år |
| **EDDA-2014-15** | Anne Birgitte Rønning | I skyggen av kanon. Empiri som utfordring i feministisk litteraturvitenskap1 |
| **EDDA-2014-16** | Anna Watz | Njutningens problematik: postfeminism, normativitet och »mommy porn» |
| **EDDA-2014-17** | Ellen Mortensen | Perler, epler og sopp: Kjønn, seksualitet og poetiske tilblivelser i Jenny Hvals Perlebryggeriet |

## Appendix B

Move structures and cycles by journal and volume, listed in descending order of number of occurrences.

**Table B1.** NPT 1994.

| | **No of Articles** | **%** |
|---|---|---|
| **Move structures and cycles:** | | |
| 1 | 12 | 42.85 |
| 1-3 | 9 | 32.14 |
| 1-2-3 | 2 | 7.14 |
| Other | 2 | 7.14 |
| 1-2-1-3-1 | 1 | 3.57 |
| 1-3-1 | 1 | 3.57 |
| 1-2-3-1 | 1 | 3.57 |
| **Total number of articles:** | **28** | |

**Table B2.** NPT 2014.

|  | No of Articles | % |
|---|---|---|
| **Move structures and cycles:** |  |  |
| 1-3 | 11 | 31.43 |
| 1-2-3 | 8 | 22.86 |
| 1 | 4 | 11.43 |
| 1-3-1-3 | 3 | 8.57 |
| 3-1-3-1-3 | 2 | 5.71 |
| 1-3-2 | 1 | 2.86 |
| 3-1-3 | 1 | 2.86 |
| 1-2-1-2-3 | 1 | 2.86 |
| 1-3-1-3-1 | 1 | 2.86 |
| 3-2-1 | 1 | 2.86 |
| 3-1-3-1-3-1 | 1 | 2.86 |
| 1-2-3-1-2-3-1-2-3-2-3 | 1 | 2.86 |
| **Total number of articles:** | **35** |  |

**Table B3.** SID 1994.

|  | No of Articles | % |
|---|---|---|
| **Move structures and cycles:** |  |  |
| 1-3 | 7 | 46.67 |
| 1-2-3 | 3 | 20 |
| 3-1-3 | 1 | 6.67 |
| 1 | 1 | 6.67 |
| 1-3-1-3-1-3 | 1 | 6.67 |
| 1-3-2-3 | 1 | 6.67 |
| Other | 1 | 6.67 |
| **Total number of articles:** | **15** |  |

**Table B4.** SID 2014.

|  | No of Articles | % |
|---|---|---|
| **Move structures and cycles:** |  |  |
| 1-3 | 6 | 42.86 |
| 1-2-3 | 2 | 14.9 |
| 1-3-2-1-3-1-3 | 1 | 7.14 |
| 1-2-1-3-1-3 | 1 | 7.14 |
| 1-3-2 | 1 | 7.14 |
| 1-3-2-3 | 1 | 7.14 |
| 1-2-3-1-2 | 1 | 7.14 |
| 1-2-3-1-2-1-2-3 | 1 | 7.14 |
| **Total number of articles:** | **14** |  |

**Table B5.** EDDA 1994.

|  | No of Articles | % |
|---|---|---|
| **Move structures and cycles:** |  |  |
| 1-3 | 8 | 29.63 |
| 1 | 6 | 22.22 |
| 1-3-2-3 | 2 | 7.41 |
| Other | 2 | 7.41 |

| 1-3-1 | 1 | 3.7 |
|:---:|:---:|:---:|
| 3 | 1 | 3.7 |
| 3-1 | 1 | 3.7 |
| 2-1-3 | 1 | 3.7 |
| 1-3-2-1-3-1 | 1 | 3.7 |
| 3-1-2 | 1 | 3.7 |
| 3-1-3-1 | 1 | 3.7 |
| 1-2-1-3-1 | 1 | 3,7 |
| 1-2-1-2 | 1 | 3.7 |
| **Total number of articles:** | **27** | 100 |

**Table B6.** EDDA 2014.

|  | **No of Articles** | **%** |
|:---|:---:|:---:|
| **Move structures and cycles:** | | |
| 1-3-1-3 | 2 | 11.76 |
| 1-3-2 | 2 | 11.76 |
| 1-3 | 2 | 11.76 |
| 1-3-2-3 | 1 | 5.88 |
| 1-2-3-1-3 | 1 | 5.88 |
| 1-2-3 | 1 | 5.88 |
| 1-2-1-2-1-3 | 1 | 5.88 |
| 1-3-2-1-2-1-3 | 1 | 5.88 |
| 1-2-1-3 | 1 | 5.88 |
| 3-2-3 | 1 | 5.88 |
| 1-3-1 | 1 | 5.88 |
| 1-3-2-1-3-2 | 1 | 5.88 |
| 1-3-1-2-1 | 1 | 5.88 |
| 1 | 1 | 5.88 |
| **Total number of articles:** | **17** | |

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
