# Peer review of "Multilingual Research Writing beyond English: The Case of Norwegian Academic Discourse in an Era of Multilingual Publication Practices"

_publications, doi:10.3390/publications7020025_

Round 1

Reviewer 1 Report

The manuscript “Multilingual Research Writing beyond English: The Case of Norwegian Academic Discourse in an Era of Multilingual Publication Practices” deals with a timely topic of multilingual author practices in academic writing. Therefore, the article has the potential attractiveness for the journal’s audience. The main goal of the manuscript is clear: the authors want to show that multilingual authors tend to adopt the rhetoric of Anglo-American academic texts when writing in other languages due to the dominance of English in academic publishing.  While I completely agree with the main argument, it seems to me that the data and the discussion do not necessarily support it.

For the purpose of the study, the authors analyzed the introductions of articles published in three Norwegian journals in the fields of education, social science, and literature in 1994 and 2014. The comparison of the articles in 1994 and 2014 showed that the articles in 2014 tended to include more of the moves given in Swales’ rhetorical model. The authors use this as evidence of English influence on multilingual writers’ texts. However, several questions come to mind regarding the evidence provided in the study.

1.      Why are the authors of the journal articles assumed to be multilingual? Have they also published in English? Have more authors from 2014 than those from 1994 published in English? Do they speak more languages than one Scandinavian language and English? In other words, how do you operationalize “multilingual”?

2.      How do we know that the possible influence comes from English? Can’t we assume that the influence comes from the greater competition and research and the need to show how your work stands out (hence explicit statement of niche)? Can’t we also assume that the types of research done, and therefore the reports, in these fields have changed over time?

3.      What is the rhetorical model of introductions in Norwegian or the other Scandinavian languages? It would be useful to know what the Scandinavian model is so that we can see how it has changed, or to know what to do in order not to maintain using the Norwegian model instead of shifting to the Anglo-American.  

4.      In the introduction, the authors mention the “parallel language use” policy. What is the specific role of the policy in relation to the current study?

It would be helpful if the authors address these questions in the manuscript in order to strengthen their arguments.

Author Response

Dear reviewer #1,

Please see the attached Word document for our response. 

Sincerely,

Kristin Solli & Ingjerd Legreid Ødemark

Reviewer 2 Report

This manuscript, as it is clearly explained in the abstract and introduction, focuses on possible changes in writing conventions in another language other than English during a period of multilingual publication patterns. It is well-written, well documented and well structured. Therefore, it is easy to follow and a pleasure to read. I highly recommend its publication.

Author Response

Dear Reviewer #2,

We thank reviewer #2 for such a positive and encouraging response. We are delighted that the reviewer found our work interesting and suitable for the journal. We hope the reviewer finds that the revised manuscript has been strengthened further.

Sincerely,

Kristin Solli & Ingjerd Legreid Ødemark

Round 2

Reviewer 1 Report

The article has been improved greatly because the authors acknowledge the limitations of the study and avoid generalizations based on the findings. Except for minor edits and formatting, the manuscript seems ready for publications.